# Imaging *Leishmania major* Antigens in Experimentally Infected Macrophages and Dermal Scrapings from Cutaneous Leishmaniasis Lesions in Tunisia

**DOI:** 10.3390/microorganisms10061157

**Published:** 2022-06-04

**Authors:** Nasreddine Saïdi, Yousr Galaï, Meriem Ben-Abid, Thouraya Boussoffara, Ines Ben-Sghaier, Karim Aoun, Aïda Bouratbine

**Affiliations:** 1Laboratoire de Recherche, Parasitoses Médicales, Biotechnologies et Biomolécules, LR 20-IPT-06, Institut Pasteur de Tunis, Université Tunis El-Manar, Tunis 1002, Tunisia; nasreddine.saidi@pasteur.utm.tn (N.S.); yousr.galai@pasteur.tn (Y.G.); meriembenabid@gmail.com (M.B.-A.); ines.bensghaier@gmail.com (I.B.-S.); karim.aoun@pasteur.tn (K.A.); 2Laboratoire de Recherche, Transmission, Contrôle et Immunobiologie des Infections, LR 20-IPT-02, Institut Pasteur de Tunis, Université Tunis El-Manar, Tunis 1002, Tunisia; boussoffara_thouraya@yahoo.fr; 3Service de Parasitologie-Mycologie, Institut Pasteur de Tunis, Tunis 1002, Tunisia

**Keywords:** cutaneous leishmaniasis, *Leishmania major*, diagnosis, dermal scrapings, microscopy, immunofluorescence assay, *Leishmania* antigen

## Abstract

*Leishmania major* cutaneous leishmaniasis (CL) lesions are characterized by an intense process of parasite destruction and antigen processing that could limit microscopic amastigote detection. The aim of our study was to develop a direct immunofluorescence (DIF) assay for in situ visualization of *L. major* antigens and access its reliability in the routine diagnosis of CL. The developed DIF assay used IgG polyclonal antibodies produced in rabbits by intravenous injections of live *L*. *major* metacyclic promastigotes chemically coupled to fluorescein isothiocyanate. Applied to *L. major* infected RAW macrophages, corresponding macrophage-derived amastigotes and dermal scrapings from CL lesions, the immunofluorescence assay stained specifically *Leishmania* amastigotes and showed a diffuse *Leishmania* antigen deposit into cytoplasm of phagocytic cells. Reliability of DIF in CL diagnosis was assessed on 101 methanol-fixed dermal smears from 59 positive and 42 negative CL lesions diagnosed by direct microscopy and/or kDNA real-time PCR. Sensitivity and specificity of DIF was 98.3% and 100%, respectively, being more sensitive than microscopy (*p* < 0.001) and as sensitive as ITS1-PCR. ITS1-PCR-RFLP allowed *Leishmania* species identification in 56 out of the 58 DIF-positive smears, identifying 52 *L. major*, two *L. infantum* and two *L. tropica* cases, which indicates antigenic cross-reactivity between *Leishmania* species.

## 1. Introduction

Cutaneous leishmaniasis (CL) is caused by a variety of *Leishmania (L.)* species transmitted to humans by the bite of phlebotomine sandflies [1,2]. It presents as skin lesions on exposed parts of the body, leaving life-long scars and causing disfigurement and distress [1,2]. About 95% of CL cases occur in the Americas, the Mediterranean basin, the Middle East, and Central Asia, with an estimated incidence between 600,000 and 1 million new cases occurring worldwide annually [1]. In North African countries, the burden of the disease is high and three *Leishmania* species, associated to distinct eco-epidemiological patterns, namely *L. infantum*, *L. major* and *L. tropica,* are involved in *Leishmania* transmission [3,4,5]. However, *L. major* is by far the most frequent species, with more than 90% of registered cases in Algeria and Tunisia [5,6].

Cutaneous leishmaniasis caused by *L. major*, also known as zoonotic or rural zoonotic cutaneous leishmaniasis (ZCL), is a major public health problem in the North African region, including Tunisia, with thousands of cases occurring each year [5,6]. It is distributed in the arid and Saharan bioclimatic stages, where it follows an epidemic pattern with seasonal occurrence of cases [5,6]. Typical ZCL lesions evolve from papules to nodules to ulcerative lesions, with a central depression and a raised, indurate border [5]. They are typically multiple and located on limbs and tend to be exudative or “wet”, large and complicated by superficial and secondary bacterial infections [5]. Most cases do not cause any clinical diagnostic difficulties, diagnosis often being suspected during epidemics on evocative clinical presentations in patients living in or coming from endemic areas [5,6,7]. However, occasionally patients present with unusual morphological forms of CL, which initially may elude diagnosis [8]. In Tunisia, parasitological confirmation of the diagnosis is highly recommended before engaging treatment [9]. It is mainly based on the direct microscopic identification of *Leishmania* amastigotes in Giemsa-stained dermal scrapings [10].

The diagnosis of CL is based on clinical features (supported by epidemiologic data) and laboratory testing. Numerous diagnostic methods have been described, including direct parasitological examination, molecular and immunological diagnostics [11,12]. Parasitological diagnosis, which is typically undertaken by direct microscopy, histopathology or culture of material from suspected lesions (obtained by scraping, needle aspiration, punch or biopsy), is still considered the gold standard in CL diagnosis because of its high specificity [11,12]. However, while these techniques are highly specific for diagnosing leishmaniasis, they are not sensitive enough [11,12]. Thus, the percent success for microscopic detection of *Leishmania* amastigotes in stained dermal scrapings varies depending on the number of parasites present and microscopist expertise, and is estimated around 60–80% for CL caused by *L. major* [6,10]. Likewise, culture on NNN medium is a less sensitive technique, which is moreover limited by the nonexceptional bacterial and fungal contamination [11,12]. To overcome these limits molecular diagnostic tests have been developed over the last decades, as these are assumed to have better sensitivity than traditional diagnostic methods [11,12]. In particular, PCRs, using either genomic or kinetoplast DNA (kDNA) and performed either as a single test or in a nested format or as a quantitative assay (qPCR), have been widely exploited [11,12]. Among these, PCR targeting kDNA minicircle is considered to be the most sensitive method for CL diagnosis, since there are about 10,000 copies of minicircles per parasite [13,14], whereas PCR assay amplifying the internal transcribed spacer 1 (ITS1) region of the rRNA genes has been shown to be a sensitive method that allows identification of almost all pathogenic Old World *Leishmania* species by restriction fragment length polymorphism (RFLP) [14]. These techniques are, however, only available in some specialized centers. Immunologic diagnostic methods are based on the detection of anti-*Leishmania* antibodies or *Leishmania* antigens. Although serologic tests are available for CL, they are not widely employed for CL diagnosis [11,12]. The CL Detect™ Rapid Test (InBios, Washington, DC, USA) targeting the peroxidoxin antigen produced by *Leishmania* amastigotes in skin lesions has been evaluated in various endemic settings with varying results [15,16]. On the other hand, it is now recognized that immunohistochemistry (IHC) detecting *Leishmania* antigens in tissue sections is a reliable complementary tool improving sensitivity and specificity of the histopathological diagnosis of CL, especially for lesions with low parasite burden [17,18]. Monoclonal [19,20] and polyclonal antibodies [18,21,22,23,24,25] produced against *Leishmania*, as well as immune serum from dog naturally infected with *Leishmania* [26], were successfully used to detect *Leishmania* amastigotes and their antigens in routinely prepared histological sections. Though antibodies detecting *Leishmania* antigens were widely used in pathology diagnosis, they were applied only in the Americas on large series of noninvasive samples [27]. Their use on dermal scrapings from Old World CL remains rarely reported [28]. There is no information available about reliability of immunofluorescence assays to access diagnosis of Old World CL in comparison to other available methods.

The aim of our study was to develop a direct immunofluorescence (DIF) assay for in situ visualization of *L. major* antigens, to examine its specificity on amastigotes-infected macrophages and macrophage-derived amastigotes, and to assess its reliability on dermal scrapings in the routine diagnosis of CL in Tunisia.

## 2. Materials and Methods

### 2.1. Leishmania Isolates and Culture

All experiments were carried out using two *Leishmania* isolates GLC94 (MHOM/TN95/GLC94) and LV59 (MHOH/TN07/LV59) collected from cutaneous and visceral cases and typed as *L. major* and *L. infantum*, respectively. These two isolates were available from the biobank of *Leishmania* species at Institute Pasteur of Tunis.

Cryopreserved *Leishmania* promastigotes were stabilized and cultured in RPMI 1640 medium (Lonza, Basel, Switzerland) supplemented with 10% heat-inactivated fetal bovine serum (FBS) (Lonza), 2 mM L-glutamine, 100 U/mL penicillin, and 100 mg/mL streptomycin. The RPMI culture was incubated at 26 °C and maintained by adding medium every 72 h. Metacyclic promastigotes were harvested during the stationary phase of culture after 5 days of incubation without culture media addition and purified through a discontinuous Ficoll gradient, as previously described [29].

### 2.2. Rabbit Anti-L. major Immune Serum

Anti-*Leishmania* antibodies were produced in 3-month-old Californian female rabbits by 5 biweekly intravenous injections of 10^8^ living *L. major* metacyclic promastigotes, as previously described [30].

### 2.3. Reactivity of the Rabbit Anti-L. major Immune Serum against In Vitro Infected Macrophages and Macrophage-Derived Amastigotes

#### 2.3.1. In Vitro Infection of Macrophages by *Leishmania* Promastigotes

The murine macrophage-like cell line RAW 264.7 (ATCC^®^ TIB-71^™^, Manassas, VA, USA) was used for in vitro infection by *L. major* metacyclic promastigote forms. Briefly, RAW 264.7 cells in RPMI 1640 medium (Sigma Aldrich, St. Quentin Fallavier, France), 100 mM sodium pyruvate, 25 mM HEPES, 100 U/mL penicillin (Sigma Aldrich), 100 μg/mL streptomycin (Sigma Aldrich), 100 mM nonessential amino acids (Sigma Aldrich) and 10% FBS were plated on a Permanox^®^ slide (10^5^ cells per well) in an 8-well chamber slide system (Nunc^®^ Lab-Tek, Sigma Aldrich) and allowed to adhere to the slides for 3 h at 37 °C, 5% CO_2_. Then, adherent macrophages were infected with *Leishmania* metacyclic promastigotes at a macrophages-to-parasite ratio of 1:10 overnight at 37 °C, 5% CO_2_. A control well of RAW 264.7 cells without parasites was set up. After overnight incubation, noninternalized promastigotes were removed by washing two times with RPMI, then culture medium was added, and infected macrophages were maintained at 37 ° C in 5% CO_2_. After two days of incubation, *Leishmania*-amastigotes-infected macrophages were visualized by microscopic examination of Giemsa-stained slides (data not shown). This two day incubation period was used for all subsequent experiments.

Amastigotes were released from infected cells by using 0.05% SDS in PBS, as previously described by Jain et al. [31]. Briefly, *Leishmania*-amastigotes-infected macrophages were washed with PBS and then exposed to 1 mL of 0.05% SDS in PBS for 30 s. The total volume of parasite suspension was aspirated, immediately diluted by adding PBS and washed 3 times to remove SDS. Amastigotes were collected by centrifugation at 2500 rpm for 10 min and resuspended in serum saline 0.9%.

#### 2.3.2. Reactivity of the Rabbit Anti-*L. major* Immune Serum

An indirect immunofluorescence (IF) assay using the rabbit anti-*L. major* immune serum followed by fluorescein-conjugated anti-rabbit IgG was tested on (i) *L. major* amastigotes-infected macrophages and (ii) corresponding derived amastigotes. Briefly, *Leishmania*-amastigotes-infected macrophages were washed twice with RPMI, fixed with formaldehyde 3% for 15 min, permeabilized with PBS 0.2% Triton X-100 for 5 min and blocked with PBS 0.5% Tween-20 3% FBS for one hour at room temperature (RT), whereas macrophage-derived free amastigotes were spotted on a slide and methanol fixed. Indirect IF used 20 µL of rabbit anti-*Leishmania* immune serum at 1/10 dilution for one hour at RT in darkness and fluorescein-conjugated anti-rabbit IgG (Invitrogen, Thermo Fisher) at 1/100 dilution for 30 min. Visualization was carried out by a Leica DM 5500, epi-illumination fluorescence microscope. Two kinds of negative controls were used in the experiments: (i) uninfected macrophages that were subjected to the entire immunofluorescence assay and (ii) infected macrophages and free amastigotes incubated with pre-immune rabbit sera as no primary antibody controls.

### 2.4. Optimization of Direct Immunofluorescence Assay (DIF) on Dermal Scrapings

Anti-*L. major* IgGs were purified from rabbit immune serum using protein A-Sepharose Column^®^ (abcam, Cambridge, UK then concentrated through an Amicon Ultra-4 filter (Millipore, Dramstadt, Germany) and quantified using BiCinchoninic acid (BC) assay (BC assay, Thermo Fischer). Purified IgGs were labeled to fluorescein isothiocyanate (FITC), as previously described [32]. Briefly, 0.2 volume of sodium carbonate buffer was added to the IgG solution to bring the pH to 9.0. Fifty microliters of FITC (Merck Millipore, Molsheim, France) at 1 mg/mL in dimethyl sulfoxide (DMSO) was prepared and added progressively to 1 mL of IgG (about 50 µg of FITC/3.2 mg IgG). The solution was incubated at +4 °C for 8 h with gentle rotation. Centrifugation through an Amicon Ultra-4 centrifugal Filter unit (Merck Millipore, Darmstadt, Germany) was used to remove unbound FITC and to concentrate the FITC-labeled IgG solution. Absorbance was read at 280 nm and 495 nm using a spectrofluorimeter. Fluorochrome-to-protein optical densities ratio (F/P) corresponded to 1+. The conjugate was stored in aliquots at −20 °C.

Methanol-fixed dermal smears from positive and negative controls were incubated with 0.2% Triton X-100 in PBS for 5 min, washed three times with PBS and incubated with 0.5% Tween-20 FBS 5% in PBS for 15 min at RT. DIF assay used 20 µL of FITC-labeled anti-*L. major* IgG at 1:100 dilution for one hour in the dark at RT. The slides were counterstained with Evans’s blue (Invitrogen) and mounted in Dako mounting fluorescence medium, then observed at 40× magnification under epifluorescence microscopy (LeicaDM5500 capture station) and LSM880 confocal microscope (Carl Zeiss Microscopy GmbH). The DIF assay was conducted with and without using DAPI (Sigma Aldrich, St. Louis, MO, USA).

### 2.5. Evaluation of DIF Assay in the Diagnosis of CL and Comparison to ITS1-PCR

Direct immunofluorescence assay and ITS1-PCR were performed on 101 methanol-fixed dermal smears from 59 confirmed CL lesions and 42 negative ones (Table 1). All slides were provided by the laboratory of Parasitology-Mycology, Pasteur Institute of Tunis and were leftover unstained slides prepared in the setting of routine diagnosis from 101 suspected CL patients. Diagnostic results from laboratory of Parasitology-Mycology, Pasteur Institute of Tunis are given on the basis of a combined reference test of microscopy and real-time kDNA PCR, which is considered positive if one of the two tests is positive, and negative if the two tests are negative, direct microscopy being systematically conducted and kDNA PCR being performed only in case of negativity of microscopy (Table 1) [10].

The DIF assay was performed as previously described without adding DAPI. The slides were counterstained with Evans’s blue in saline and mounted with buffered glycerol (pH 8.0) and observed at 40× magnification under epi-fluorescence microscopy (LeicaDM5500 capture station). After DIF assay, glycerol was wiped off with absolute ethanol and material forming the dry smear was scraped with a sterile scalpel and covered with 200 µL Qiagen Lysis buffer (Qiamp DNA Blood Mini Kit, Qiagen, Hilden, Germany) for 5 to 10 min before transfer to a 1.5 mL reaction tube. Enzymatic digestion was performed overnight with proteinase K, then DNA was extracted according to the manufacturer’s recommendations. DNA elution was conducted in 50 µL AE buffer and used for ITS1 PCR, as previously described [33]. *Leishmania* species identification was achieved using restriction profile analysis in comparison to *L. major*, *L. tropica* and *L. infantum* reference isolates [34]. Parasitic load of dermal smears from microscopy-negative/kDNA qPCR-positive CL lesions was estimated by kDNA qPCR performed, as previously described by Mary et al. using a standard curve (from 10^4^ to 0.01 parasites/µL) generated from a dilution series of *Leishmania* DNA extracted from 10^6^
*L. infantum* promastigotes [35].

### 2.6. Statistical Analysis

Sensitivity and specificity of DIF assay and ITS1-PCR were computed considering results of routine diagnosis as a gold standard. Chi-squared test was used for comparison of proportions. The significance level was set at 5%.

### 2.7. Ethical Considerations

All experimentations on animals were approved by the Animal Ethics Committee of the Pasteur Institute of Tunis, Tunisia (Reference 2016/08/I/LR11IPT06/V3, 22 March 2019). Methanol-fixed slide smears were prepared from dermal scraping in the setting of routine diagnosis of CL and correspond to residual unstained slides. Slide smears were anonymized and tests were performed in blinded conditions. This work was carried out in accordance with the relevant guidelines and regulations, and does not provide any personal data.

## 3. Results

### 3.1. Reactivity of the Rabbit Anti-L. major Immune Serum against In Vitro Infected Macrophages and Macrophage-Derived Amastigotes

The indirect IF test using rabbit anti-*L. major* immune serum was positive when applied on in vitro infected macrophages or on amastigotes (Figure 1C,D). In contrast, in control experiments, (i) uninfected macrophages did not react with rabbit anti-*L. major* immune serum and (ii) infected macrophages did not react with the preimmunization rabbit serum (Figure 1A,B).

*Leishmania major*-infected RAW cells showed different fluorescence images (Figure 1C). The most representative was a dense speckled cytoplasmic fluorescence suggesting the presence of *Leishmania*-derived antigens into cytoplasm of infected host cells. Some RAW cells demonstrated brilliant surface fluorescence labeling and several amastigotes were revealed outside macrophages as fluorescent stained bodies.

### 3.2. Detection of Leishmania Antigens by Direct Immunofluorescence Staining of Dermal Scrapings

In dermal scrapings from confirmed *L. major* CL cases, the FITC-labeled anti-*L. major* IgG detected infected mononuclear phagocytic cells and *Leishmania* amastigotes (Figure 2A–C), whereas, in negative controls, no fluorescence was shown (Figure 2D). Infected mononuclear phagocytic cells were identified by a fluorescence cell pattern with *Leishmania* antigen appearing as a diffuse deposit into the cytoplasm (Figure 2A–C). Amastigotes were identified by their size and shape into phagocytes (Figure 2A,C).

### 3.3. Reliability of Direct Immunofluorescence (DIF) Method Using FITC-Labeled Anti-L. major IgG in Comparison to Other Available Methods

The DIF assay was positive in 58 out 59 positive CL cases and negative in all negative ones, giving a sensitivity of 98.3% and a specificity of 100% in CL diagnosis. The DIF assay results 100% correlated with those of ITS 1-PCR. However, DIF was more sensitive than direct microscopy (98.3% versus 71.2%, *p* < 0.001) (Table 1).

The DIF assay allowed diagnosis for 16 out of the 17 smears from CL lesions negative by microscopy and positive by kDNA qPCR. Parasite quantification from these latter slides was globally low with a median parasitic load of 60 parasites/smear and an interquartile range of 13–123 parasites/smear. The only negative slide by DIF assay had a very low parasitic load, which was estimated at four parasites/smear.

ITS1 PCR-RFLP analysis of DNA extracted from the DIF-positive slides allowed species identification in 56 cases out of 58. These were 52 cases of *L. major*, two cases of *L. infantum* and two cases of *L. tropica* (Table 1).

## 4. Discussion

Rabbit intravenous infection with live *Leishmania* metacyclic promastigotes was already used as a cheap and non-laborious way to produce anti-*Leishmania* amastigote antibodies [36,37]. Used in an indirect immunofluorescence assay, our anti-*L. major* serum demonstrated positive staining of *L. major* amastigotes released from infected RAW cells. On the other hand, we demonstrated in a previous study that these polyclonal IgG antibodies recognize the promastigote form of *Leishmania* and cross-react with other *Leishmania* species than *L. major* [30].

Applied on in vitro *L. major*-infected RAW macrophages the anti-*L. major* serum followed by FITC-conjugated goat anti-rabbit IgG showed fluorescence images suggestive of the presence of *Leishmania*-derived antigens into the cytoplasm and on the membrane of the host cells. The key to providing *Leishmania* antibody specificity in these experiments was the use of two kinds of negative controls, ensuring that uninfected macrophages are not recognized by the polyclonal antibodies and that fluorescence staining is produced only when the anti-*L. major* antibodies are used. The observed fluorescence images are in concordance with results of previous experimental studies, indicating that *Leishmania* infection of macrophages is followed by the appearance of *Leishmania* antigen from amastigote origin on the external surface membrane of macrophages [38,39]. Furthermore, other authors evaluated the distribution of *Leishmania* proteins in infected RAW cells and showed that some of these are secreted into parasitophorous vacuoles (PVs) and then traffic out PVs into the host cell cytosol and nucleus in vesicles of distinct morphologies [40].

Anti-*Leishmania* IgGs were purified and coupled to FITC. The latter remains the most commonly used amine-reactive fluorophore for fluorescence labeling, with an excellent fluorescence potential and with high-quality images [41]. Moreover, DIF assay offers the advantage over the indirect immunofluorescence procedure of reducing nonspecific background signal and limiting the possibility of antibody cross-reactivity through the use of conjugated primary antibodies. On dermal scrapings, the DIF assay was optimized using DAPI, a cell nuclear-specific dye. DAPI blue fluorescence allowed marked contrast with the green fluorescence of probed antigens and, more importantly, allowed localization of the cells present in the dermal scraping via their nuclei. This made it easier to visualize in dermal scrapings from confirmed *L. major* CL cases, mononuclear cells with intracytoplasmic fluorescent green structures, with size and shape corresponding to those of *Leishmania* amastigotes, but also fluorescent deposits likely corresponding to *Leishmania*-derived antigens. The presence of this intracytoplasmic leishmanian material has been previously documented by IHC on biopsies from *L. tropica* and *L. braziliensis* CL lesions [21,24]. Another IHC study has documented the intracellular presence of amastigote degradation products and soluble degradation components [41]. For CL, due to *L. major*, intracytoplasmic leishmanian material existence into macrophages is likely related to a dynamic and intense process of amastigote destruction and subsequent antigen processing [42,43].

Applied on dermal scrapings, the DIF assay was more sensitive than microscopy and as sensitive as ITS1 PCR. Furthermore, the gain in sensitivity of DIF and ITS1-PCR was achieved mainly in smears containing about 10–100 parasites. Our previous studies (carried out in the same epidemiological setting) already reported that this low parasite load was not detected by microscopy but was by conventional ITS1 PCR, a lower load not being detected by either technique [10]. This high sensitivity of the DIF assay, which reaches that of the molecular tool, corroborates results obtained with histopathological diagnosis [17,18,23], polyclonal antibodies produced against *Leishmania* showing higher *Leishmania* detection rate compared to monoclonal ones [26]. Thus when the parasite load is low and ordinary techniques such as Giemsa stains fail to detect the parasites, immunohistochemistry with anti-*Leishmania* antibodies has demonstrated a higher level of sensitivity in the identification of amastigotes [44,45]. Furthermore, in our study, most CL cases were caused by *L. major* and DIF assay stained, in addition to the parasites, amastigote-derived antigens into host cells, which may have enhanced sensitivity of the method.

On the other hand, the polyclonal anti-*L. major* IgGs enabled detection of *L. infantum* (*n* = 2) and *L. tropica* (*n* = 2) cases, which also explain their high sensitivity in detecting *Leishmania*. This result is in accordance with those of previous studies indicating antigenic cross-reactivity between *Leishmania* species. Thus, immune serum from dog naturally infected with *L. chagasi* was successfully used to diagnose American tegumentary leishmaniasis [26]. Moreover, no difference in fluoresecence staining was reported with antisera to *L. tropica*, *L. major* or *L. Mexicana* [41]. Further complementary study is, however, needed to better evaluate the test accuracy according to species. On the other hand, although the negative controls validated the specificity of polyclonal antibodies against *Leishmania* antigens in the experimental study, and the DIF assay applied to negative dermal scrapings did not visualize parasites and/or infected cells, the specificity of DIF in the diagnosis of CL needs to be evaluated by a dedicated protocol. The latter should include specimens of other infectious and noninfectious skin diseases whose lesions are clinically suggestive of CL [8,9].

## 5. Conclusions

*Leishmania major* is the main *Leishmania* species causing CL in Tunisia. Its lesions are characterized by an intense process of parasite destruction and antigen processing that could limit microscopic amastigote detection and delay patient management. Here, we described the development of a direct immunofluorescence (DIF) method that stains specifically, in addition to *Leishmania* amastigotes, their degraded residues and soluble components. The DIF assay applied to dermal scrapings showed fluorescence images suggestive of both parasites and infected cells and was more sensitive than microscopy and as sensitive as ITS1-PCR. Furthermore, the DIF assay using anti-*L. major* polyclonal IgG recognized the other *Leishmania* species present in Tunisia. Used as a second-line test, DIF could improve management of patients with a false-negative microscopy result.

## Figures and Tables

**Figure 1 microorganisms-10-01157-f001:**
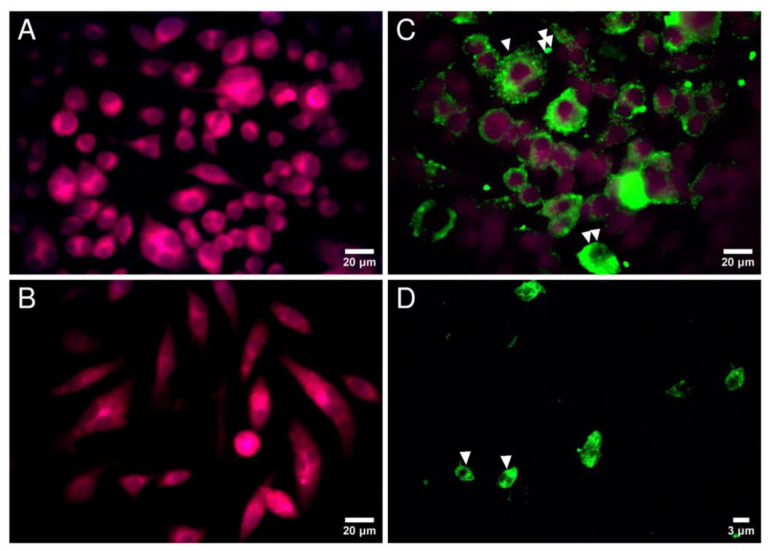
Fluorescence light micrographs of *L. major*-infected RAW 264.7 cells and corresponding macrophage-derived amastigotes treated with rabbit anti-*L. major* immune serum and FITC-conjugated goat anti-rabbit IgG. (**A**,**B**): Control experiments observed by epifluorescence microscopy (×40). A: Noninfected RAW 264.7 cells treated with rabbit anti-*L. major* immune serum and FITC-conjugated goat anti-rabbit Ig G; B: infected RAW 264.7 cells treated with preimmune serum. (**C**): *Leishmania major* infected RAW 264.7 cells treated with rabbit anti-*L. major* immune serum and FITC-conjugated goat anti-rabbit IgG and observed by epifluorescence microscopy (×40). A dense speckled cytoplasmic fluorescence (one arrow) and a brilliant surface fluorescence labeling (2 arrows) are shown with RAW cells; amastigotes of *L. major* appear as extra-cellular fluorescent bodies (3 arrows). (**D**): *L. major* macrophage-derived amastigotes treated with rabbit anti-*L. major* immune serum and FITC-conjugated goat anti-rabbit IgG and observed by epifluorescence microscopy (×100).

**Figure 2 microorganisms-10-01157-f002:**
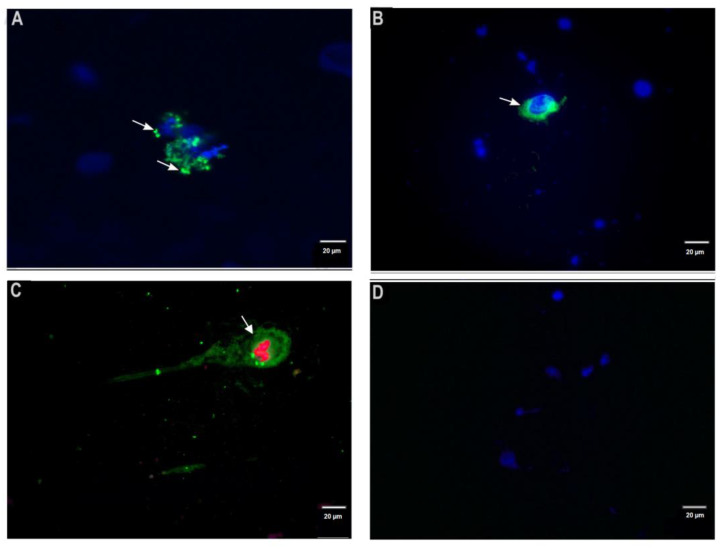
Detection of *L. major* infected cells and *L. major* amastigotes on fixed dermal scrapings slides by direct immunofluorescence assay. (**A**): The slide was observed by confocal microscopy at objective ×40. Fluorescent amastigotes (green) (arrow) were clearly shown inside host cells, which were recognized by their nucleus stained in blue by DAPI. (**B**): The slide was observed by epifluorescence microscopy at objective ×40. *Leishmania* antigen (arrow) appears as a diffuse deposit in the cytoplasm of mononuclear phagocytic cells, which were recognized by their nucleus stained in blue by DAPI. (**C**): The slide was observed by epifluorescence microscopy at objective ×40. The DIF assay was performed without DAPI. *Leishmania* antigen appears as a diffuse deposit inside the host cell (arrow), which also harbors *Leishmania* amastigotes. (**D**): The dermal scraping of suspected CL lesion with both negative microscopic examination and qPCR did not show fluorescence. Nuclear cells are clearly observed with DAPI.

**Table 1 microorganisms-10-01157-t001:** Results of direct immunofluorescence assay and ITS1 PCR-RFLP on dermal smears from suspected CL lesions.

Routine Diagnosis of Suspected CL Lesions (*n* = 101)		Corresponding Methanol Fixed Dermal Smears (*n* = 101)							
		DIFA		ITS1 PCR		ITS1 PCR-RFLP			
		Positive	Negative	Positive	Negative	*L. major*	*L. infantum*	*L. tropica*	ND *
Positive (*n* = 59)	Positive microscopy kDNA qPCR not done (*n* = 42)	42	0	42	0	38	2	2	0
	Negative microscopy Positive kDNA qPCR (*n* = 17)	16	1	16	1	14	0	0	2
Negative (*n* = 42)	Negative microscopy Negative kDNA qPCR (*n* = 42)	0	42	0	42	-	-	-	-

* Not determined.

## Data Availability

Not applicable.

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
