# Peer review of "Imaging Leishmania major Antigens in Experimentally Infected Macrophages and Dermal Scrapings from Cutaneous Leishmaniasis Lesions in Tunisia"

_microorganisms, 2022, doi:10.3390/microorganisms10061157_

Round 1
Reviewer 1 Report
The manuscript entitled “Imaging Leishmania major antigens in experimentally infected macrophages and dermal scrapings from cutaneous leishmaniasis lesions in Tunisia” developed a direct immunofluorescence (DIF) assay for in situ visualization of L. major antigens and access its reliability in the routine diagnosis of CL. The manuscript is well written, and should be of great interest to the readers. However, the manuscript needs improvement regarding the background information. Different modifications are necessary:
-write “Leishmania, L. major, L. tropica and infantum in italic;
-Line 18 replace “L.major” with “L. major”
-Line 59- add bibliography
-Several methods and tools have been developed over recent years for the detection, quantification, and identification of the parasite of the genus Leishmania. It would be necessary to understand the amount of parasitic charge in the samples under study. The results section does not include parasite quantity data (qPCR), please include all available results.
- increase with more updated bibliography the section "discussions", trying to solve the diagnostic problems due to the low sensitivity of microscopic examination or the low predictive values of serology, whose results can be affected by either persistent antibodies (false positive) or immunosuppression (false negative), inserting the limitations of the method. Explain the discussion in more detail.
-“in vitro” should be written in italic.
Finally, once these changes are done, I recommend that the paper should be accepted for the publication in this journal.
Author Response
Attached is the manuscript revised along the lines recommended. All comments have been taken into account.
Please find below an itemized list of specific answers to each of your comments, with notification where changes have been made in the revision. We hope that these corrections will be satisfactory.
1- Introduction:
-the manuscript needs improvement regarding the background information.
-write “Leishmania, L. major, L. tropica and infantum in italic;
-Line 18 replace “L.major” with “L. major”
-Line 59 add bibliography
Answer: The introduction was fully reviewed and corrected (lines 34 – 103). bibliography was updated
2- Methods/results
-Several methods and tools have been developed over recent years for the detection, quantification, and identification of the parasite of the genus Leishmania. It would be necessary to understand the amount of parasitic charge in the samples under study. The results section does not include parasite quantity data (qPCR), please include all available results.
Answer: Information about parasite quantification was added (lines 203 – 208) and (lines 276 -280)
3- Discussion
- increase with more updated bibliography the section "discussions", trying to solve the diagnostic problems due to the low sensitivity of microscopic examination or the low predictive values of serology, whose results can be affected by either persistent antibodies (false positive) or immunosuppression (false negative), inserting the limitations of the method. Explain the discussion in more detail.
-“in vitro” should be written in italic.
Answer: The section dealing with low sensitivity of microscopic examination was corrected (lines 330-343). Bibliography was updated. “in vitro” was written in italic
Reviewer 2 Report
The manuscript submitted to Journal microorganisms is simply and clearly written.
In the part of introdution I would appreciate to cite newer studies or rather review articles, mainly in general, well known informations (the first sentence of introduction, please change references 1 and 2 for some recent review).
In the part of Materials and Methods I miss almost everywhere italics in Leishmania genera and species, please, correct. Further, I miss a passage nuber of parasites in chapter 2.2.
In the part of Results, scales on the picture 1 (A,B,C,D) are illegible (low quality). Further, I have a problem with table 1. I don´t understand + and - symbols in a SESPECTED CL LESION column, please, clarify. Negative dernal smear were not examined with any method? (microscopy, PCR)?
Line 276, intra-cytoplasmic?
I would appreciate to discuss more Leishmania Ab specificity and known differences among antigens of L. major, L. infantum and L. tropica. What do think about situation if you would infect rabbit with L. infantum or L. tropica. Would be the immune raction more species-specific?
I think that authors should discuss their results with another studies, e.g. Eric Muraille at al., 2010, N Basu et al., 1994.....
I don´t think that a reference number 12 is about contamination as authors mentioned in the introduction. The second paper, I can not assess, because text is in French.
Author Response
Attached is the manuscript revised along the lines recommended. All comments have been taken into account.
Please find below an itemized list of specific answers to each of your comments, with notification where changes have been made in the revision. We hope that these corrections will be satisfactory.
1- Introduction:
I would appreciate to cite newer studies or rather review articles, mainly in general, well known informations (the first sentence of introduction, please change references 1 and 2 for some recent review).
Answer: The introduction was fully reviewed and corrected (Lines 34-103). Bibliography was updated as recommended
2- Materials /Methods
In the part of Materials and Methods I miss almost everywhere italics in Leishmania genera and species, please, correct.
Answer: This was corrected
Further, I miss a passage number of parasites in chapter 2.2.
Answer:
Culture was developed in chapter 2.1. Cryopreserved Leishmania promastigotes were stabilized and cultured in RPMI 1640 medium supplemented with 10% FBS, incubated at 26°C and maintained by adding medium every 72 hours. Metacyclic promastigotes were harvested during the stationary phase of culture after 5 days incubation without culture media addition.
This was repeated for every injection.
3- Results
-In the part of Results, scales on the picture 1 (A,B,C,D) are illegible (low quality).
Answer: Quality of picture 1 was improved
-Further, I have a problem with table 1. I don´t understand + and - symbols in a SESPECTED CL LESION column, please, clarify. Negative dernal smear were not examined with any method? (microscopy, PCR)?
Answer: we clarified this in the methods (lines 187-192) and results by correcting the table
4- Discussion
Line 276, intra-cytoplasmic?
Answer: This was corrected
I would appreciate to discuss more Leishmania Ab specificity and known differences among antigens of L. major, L. infantum and L. tropica. What do think about situation if you would infect rabbit with L. infantum or L. tropica. Would be the immune raction more species-specific?
Answer: We improve the discussion (lines 344-350)
5- References
I think that authors should discuss their results with another studies, e.g. Eric Muraille at al., 2010, N Basu et al., 1994.....
I don´t think that a reference number 12 is about contamination as authors mentioned in the introduction. The second paper, I can not assess, because text is in French.
Answer: We corrected and updated references